# Hand hygiene after the COVID-19 pandemic: Is it still at a high level?

Jinqi Wang[1,2], Yaofei Liu[2,3], Neng Li[1,2], Yifen Zhu[1,2], Min Du[2,4], Xiaoman He[1,2]*

**1** Department of infection prevention and control, The Central Hospital of Wuhan, Tongji Medical College, Huazhong University of Science and Technology, Wuhan, P. R. China, **2** Key Laboratory for Molecular Diagnosis of Hubei Province, The Central Hospital of Wuhan, Tongji Medical College, Huazhong University of Science and Technology, Wuhan, P. R. China, **3** Department of laboratory medicine, The Central Hospital of Wuhan, Tongji Medical College, Huazhong University of Science and Technology, Wuhan, P. R. China, **4** Department of Clinical Nutrition, The Central Hospital of Wuhan, Tongji Medical College, Huazhong University of Science and Technology, Wuhan, P. R. China

* victory1219fly@outlook.com

**Data availability statement:** All relevant data are within the paper and its Supporting information files.

**Funding:** This study was funded by Wuhan Science Foundation for Youths [Grant number: WG21Q03]. The funder had no role in study design, data collection and analysis, decision to publish, or preparation of the manuscript.

## Abstract

### Background

Hand hygiene (HH) is crucial for preventing healthcare-associated infections (HAIs), with compliance notably increasing during the COVID-19 pandemic. Whether this adherence can be sustained remains uncertain. We sought to assess changes in HH compliance following the pandemic and explore its correlation with HAIs.

### Methods

A longitudinal study was launched to assess HH compliance and HAIs across two phases. Phase 1 spanned from October 2021 to January 2023, a period of normalized prevention and control of COVID-19, while phase 2 extended from February 2023 to July 2024, following the downgrade in the management of COVID-19. Observers recorded HH practices. HAIs were identified through using semi-automated, continuous surveillance software. Pearson correlation coefficient ($rs$) was used to evaluate the relationship between HAIs and HH.

### Results

A total of 2,233 HH opportunities were observed with 966 during Phase 1 and 1,267 in Phase 2. Compliance with HH significantly declined from 90.27% (95% CI: 85.34% to 95.4%) in Phase 1 to 82.56% (95% CI: 80.35% to 84.61%) in Phase 2 ($P < 0.001$). Concurrently, the incidence of HAI rose notably from 16.79‰ (95% CI: 16.20‰ to 17.39‰) to 18.71‰ (95% CI: 18.20‰ to 19.24‰) ($P < 0.001$). However, despite these concurrent trends, monthly analysis found no statistically significant correlation between HH compliance and HAI incidence ($rs = -0.296$; $P = 0.0897$).

**Competing interests:** The authors declare that they have no relevant financial or non-financial interests to disclose.

## Conclusions

As the pandemic threat waned, maintaining high HH compliance became challenging, indicating it's time to implement additional strategies. Moreover, the correlation between HH and HAI requires further study to uncover.

## Introduction

Hands are the main pathways of pathogen transmission and are central to the transmission of diseases within the healthcare settings [1]. Healthcare-associated infections (HAIs) pose a significant challenge to global health systems, leading to extended hospital stays and a subsequent increase in healthcare costs [2,3]. As reported by the World Health Organization (WHO), out of every 100 patients in acute-care hospitals, 7 patients in high-income countries (HICs) and 15 patients in low- and middle-income countries (LMICs) will develop at least one HAI during their hospitalization, with 10% will die due to the infection [4]. Hand hygiene (HH) is the most effective single intervention promoted for the prevention of HAIs [5,6]. A 4-year intervention on HH compliance in China showed an increase from 68.90% to 91.76% with a concurrent decrease in HAI incidence from 1.10% to 0.91% [7]. A multimodal campaign to improve HH in a gynecology/obstetrics tertiary-care center in Vietnam resulted in a compliance increase from 21.50% in 2010 to 75.10% in 2018, and a significant reduction in HAI incidence from 1.10 episodes per 1000 patient-days to 0.45 per 1000 patient-days [8].

Owing to the importance of HH, monitoring HH compliance are essential to illustrate HH compliance and to evaluate the impact of targeted interventions. HH Compliance is often assessed through self-reporting, direct observation, video auditing, and electronic monitoring systems, each with its own limitations, such as recall bias, Hawthorne effects, ethical or legal concerns, and considerable financial and temporal investments [1,9]. Until now, direct HH observation is still considered as the best method to assess HH compliance, despite the Hawthorne effect.

The COVID-19 pandemic had further emphasized the need for improvement in HH compliance by Healthcare Workers (HCWs) [10]. Seven studies with 2,377 health care providers reporting the estimated overall HH compliance during the pandemic was 74.00%, which was higher than that reported in pre-pandemic studies (5.00%−89.00%) [11]. Previous studies suggested that adequate HH behaviors and compliance discontinue over time following an epidemic or pandemic [12,13]. However, it remains uncertain whether the overall HH compliance still sustain at a high level as the focus on COVID-19 pandemic wanes.

Therefore, the aim of this study was to evaluate the HH compliance and its characteristics during the period of normalized prevention and control of COVID-19, as well as after the COVID-19 management downgrade. Additionally, the correlation between HH compliance and the incidence of HAIs was evaluated.

## Methods

### Study design/population

This longitudinal study was carried out in three campuses of a tertiary hospital from October 2021 to May 2024, covering over 4,000 HCWs. The study evaluated HH compliance among HCWs across various departments, including the medical wards, surgical wards, ICU, operating room, outpatient clinics and other units.

As the COVID-19 rapidly spread across the world, China implemented its response and control measures under the framework established by the Law of the People's Republic of China on Prevention and Treatment of Infectious Diseases (Adopted in 1989, revised in 2004, 2013 and last revised in 2025) [14], which classifies infectious diseases into Categories A, B, and C based on their severity and social impact. Although initially classified as Category B, COVID-19 was managed with Category A control measures (the strictest level) nationwide since January 2020, as mandated by China's National Health Commission [15]. This exceptional approach ("Category B disease with Category A measures") was implemented due to its pandemic potential. The hospital, situated in a key epidemiological area, enacted rigorous preventive and control tactics, including reconfigure the ward layout and setting up "three areas" and "two channels", monitoring the body temperature strictly, stringent monitoring of HH with immediate feedback and others. After more than three years' fighting against the virus, the management of COVID-19 was downgraded to a Category B infectious disease on January 8, 2023 in China [16]. This study was designed with 2 distinct phases (Table 1). Phase 1, from October 2021 to January 2023, corresponds to the period of normalized prevention and control of COVID-19, during which strict infection prevention and control measures were implemented. Phase 2, from February 2023 to July 2024, marks the period following the downgrade in the management for COVID-19.

### Observation of HH compliance

According to the WHO's strategy outlined in their 2009 guidelines [17], HH compliance is the act of either washing or disinfecting hands at an opportunity for HH. The "Five Moments for Hand Hygiene" framework, first systematically introduced in the WHO Hand Hygiene Technical Reference Manual [18], categorizes these opportunities into five critical moments: (1) Prior to patient contact, (2) Prior to a clean or aseptic procedure, (3) After contact with body fluid, (4) After patient contact, and (5) After contact with the patient environment. The assessment of HH practices was conducted through direct observation using the WHO observation form consisting of 5 moments, which has been translated into Chinese for local use. The procedures for hand rub and hand wash were defined by the WHO as follows: 'hand rub' involves the application of an antiseptic hand rub to reduce or inhibit microbial growth, while 'hand wash' refers to the cleansing of hands with soap and water.

Direct observation was used to assess HCWs' HH compliance without prior notification during working hours. To minimize the Hawthorne effect, observations were discreetly integrated into the routine tasks of observers, ensuring that the HCWs were unaware that HH practices were being observed. Ethical approval (2021−016) was obtained, which specifically waived the requirement for individual informed consent for these observations. Twelve staff members from the

**Table 1. Study overview and description.**

| Phase | Period (Months) | Description |
|---|---|---|
| **Phase 1 (Normalized prevention and control of COVID-19)** | Oct. 2021 – Jan. 2023 (16 months) | HH guidelines established for COVID-19 outbreak management including intensive training, ample provision of PPE, strict audit and immediate feedback. |
| **Phase 2 (Management for COVID-19 downgrade)** | Feb. 2023 – Jul. 2024 (18 months) | Adhering to HH guidelines established before and during the pandemic, with no emphasis on implementing additional active improvement measures. |

infection prevention and control department, proficient in the WHO's 5 Moments for HH and equipped with insights gained from case scenarios and instructional videos, served as observers.Stratified random sampling was employed to select HCWs for observation. Departments in the hospital were stratified into six categories, including surgical wards, medical wards, operating rooms, outpatient clinics, ICUs, and other units (including radiology, laboratory, rehabilitation, and other supporting departments). The proportion of total observation sessions allocated to each stratum was determined by their characteristics and HAI risks: medical wards (15%), surgical wards (15%), operating rooms (15%), outpatient clinics (20%), ICUs (20%), other units (15%). The twelve observers were divided into six fixed pairs, with each pair assigned exclusively to one of the six strata.

HCWs were categorized into doctors, nurses, paramedical staff, and cleaning staff. Specifically, doctors and nurses were primarily distributed across clinical departments (medical wards, surgical wards, operating rooms, outpatient clinics, ICUs); paramedical staff mainly in diagnostic/therapeutic departments (radiology, laboratory, rehabilitation); and cleaning staff across all departments.

Observation sessions were conducted between 7:30 and 10:30 a.m., as this period involves high-intensity care activities that yield a dense and diverse range of HH opportunities under typical workload pressures. During each session, the pre-assigned pair of observers for the stratum visited one department/ward and discreetly observed HH practices for 20–30 minutes. Within this timeframe, they recorded HH opportunities encountered at random among multiple HCWs engaged in routine care, adhering to two protocols: (1) no HCW was observed more than once within the same session, and (2) HCWs observed within the previous week were excluded from subsequent observations. Observers independently documented the concurrent care activities and HH actions on separate paper forms, indicating the HCW's profession, HH moments, action taken (rub/wash), and glove usage. These forms were later compared, with discrepancies resolved through discussion to ensure the accuracy of the observations. HH compliance was calculated as the number of HH performed divided by the number of opportunities observed. No individual feedback or intervention was provided to observed HCWs during or after the observations. However, aggregate HH compliance rates of the hospital were publicly reported quarterly through the Hospital Infection Management Committee briefing and Infection Control Bulletin to maintain transparency.

## HAI

The study focused on inpatients who developed infections during their hospital stay. Real-time monitoring of HAIs was facilitated by an Infection Surveillance Software, which integrates seamlessly with the hospital's electronic medical records, provides a comprehensive surveillance system. The software is programmed to alert physicians through a notification interface whenever a patient has a persistent fever exceeding three days or when there are abnormal results from biochemical, microbiological, and imaging studies. Upon alert receipt, physicians perform an initial assessment to check if the patient's conditions match the preliminary diagnostic criteria for HAIs as outlined by the Health Ministry of China [19]. Subsequently, infection control specialists conduct a meticulous review of the patient's medical records and confirm the HAI diagnosis.

The incidence of HAIs throughout the study period was determined by calculating the ratio of HAIs to the total inpatient population.

## Statistical analysis

Statistical analyses were performed using R software version 4.2.1 (R Foundation for Statistical Computing, Vienna, Austria), in conjunction with the 'epiDisplay' and 'Hmisc' packages. Compliance rates, both overall and stratified by subgroup, were calculated as percentages, with their respective 95% confidence intervals (95% CI) also determined. To assess differences between groups and phases, cross-tabulations and Pearson Chi-Square tests were employed. Pearson's regression model was used to detect the correlation between HH compliance and HAI incidence.

### Ethics approval and consent to participate

The observation period for this study spanned from October 2021 to July 2024. Considering the nature of the study, a waiver of informed consent was granted and approved by the Ethics Committee of The Central Hospital of Wuhan, Tongji Medical College, Huazhong University of Science and Technology (Ref no: 2021−016).

## Results

### HH compliance

A total of 2,233 HH opportunities were observed during two phases, and HCWs conducted HH according to recommendations in 1,918 of the occasions, yielding a total compliance rate of 85.89% (95% CI: 84.38% to 87.31%). In Phase 1, we identified 966 moments for HH practices. This number rose to 1,267 in Phase 2. Detailed data are in S1 Table in the Supporting Information. The HH compliance decreased significantly from 90.27% (95% CI: 85.34% to 89.54%) in Phase 1 to 82.56% (95% CI: 80.35% to 84.61%) in Phase 2 ($P < 0.001$), as depicted in S1 Table.

### HH compliance by different healthcare worker groups

Individual compliance rates for each healthcare worker group (doctor, nurse, paramedical staff, cleaning staff) are presented in Fig 1a. Excluding cleaning staff, the HH compliance rates for all other groups of HCWs were higher in Phase 1 compared to Phase 2, with the difference being statistically significant for both doctors and nurses ($P < 0.001$).

The comparison of HH compliance observed in two phases among different professions shows that nurses have a significantly higher HH compliance than doctors and paramedical staff (Fig 1b). Detailed data are in S2 Table.

### HH compliance by moments

HH Compliance was analyzed separately (Fig 2) for each of the "5 moments" for HH (M1 = Prior to patient contact, M2 = Prior to a clean or aseptic procedure, M3=After contact with body fluid, including cases where gloves were worn, M4=After patient contact, M5=After contact with the patient environment.). Except for M5, the HH compliance in Phase 1 was higher than in Phase 2 for all other moments. Among them, the HH compliance for M4 in Phase 1 was significantly higher than in Phase 2 ($P < 0.001$) (Fig 2a).

Fig 2b illustrates a comparison of HH compliance among different moments during two phases, revealing that compliance for M3 was significantly lower than other moments. After excluding instances where gloves were used during M3 (referred to as M3-1), the compliance rose to 87.50%, with no significant difference observed between M3-1 and other moments (S1 Fig). This disparity may stem from HCWs' tendency to wear gloves when in contact with body fluids, leading to a potential neglect of HH practices upon glove removal. Additionally, HH compliance for M5 was significantly higher compared to M1 and M4 ($P < 0.05$). Detailed data on different moments can be found in S3 Table.

### HH compliance by departments

Fig 3 depicts HH compliance across various departments, revealing a general decline in adherence among two phases. Notably, a significant decrease in HH compliance was observed in surgical wards, outpatient clinics, and other units ($P < 0.05$, $P < 0.001$, $P < 0.05$ respectively) (Fig 3a).

Fig 3b illustrates the comparative analysis of HH compliance across different departments, revealing that HH compliance in the Intensive Care Unit (ICU) was notably higher than in medical wards ($P < 0.001$), surgical wards ($P < 0.001$), outpatient clinics ($P < 0.05$), and other units ($P < 0.05$). Detailed data can be found in S4 Table.

### HH compliance by seasons

For all seasons, HH compliance was higher in Phase 1 than in Phase 2. Notably, the HH compliance for summer and autumn in Phase 1 was significantly greater than that in Phase 2 ($P < 0.05$) (S3a Fig).

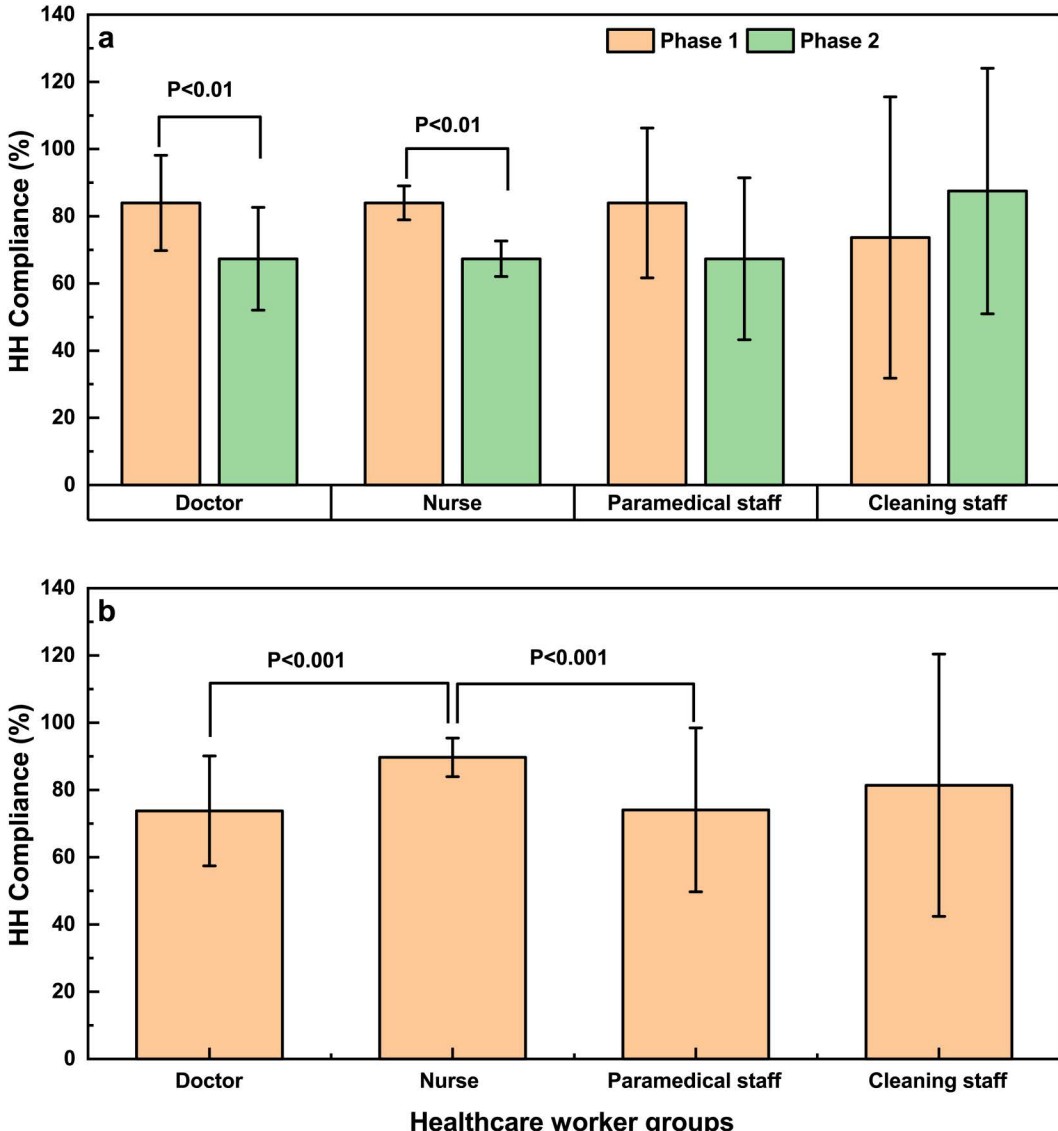

**Fig 1. Comparison of HH compliance (a) between Phase 1 and Phase 2 among different healthcare worker groups and Changes in HH compliance (b) for different healthcare worker groups.**

Comparisons of HH compliance between the two phases across different seasons revealed that HH compliance in autumn was significantly higher than in spring ($P < 0.001$), summer($P < 0.05$), and winter($P < 0.05$)(S3a Fig). Detailed seasonal HH compliance data are provided in S5 Table.

### Hand-rubbing duration>15 s

The incidence of hand-rubbing for a duration exceeding 15 s was examined across different phases and methods, including the use of alcohol-based hand rub (ABHR) and soap & water. The use of ABHR was the preferred HH action, occurring 1,319 times and representing 59.07% of all HH behaviors.

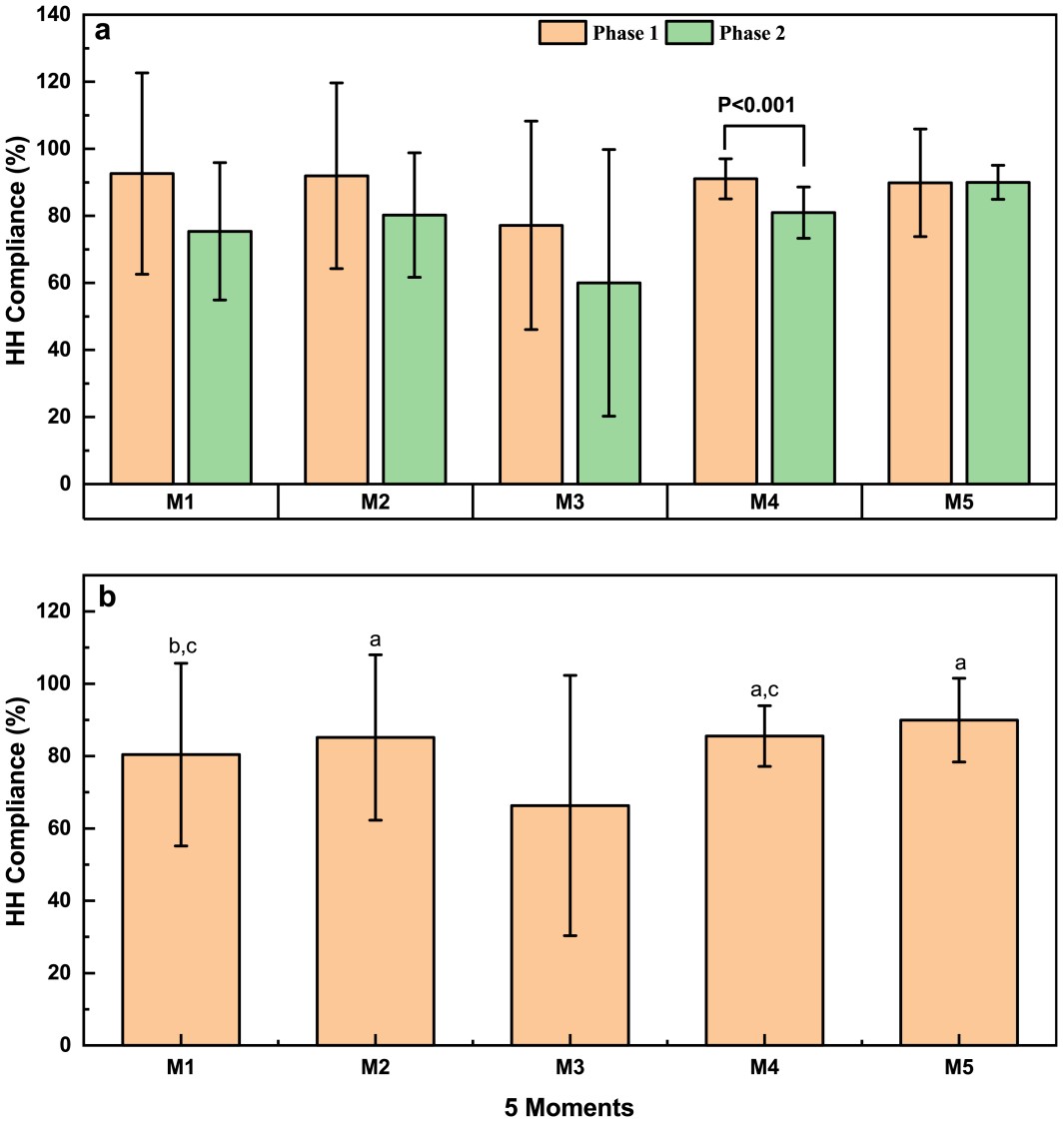

**Fig 2. Comparison of HH compliance (a) between Phase 1 and Phase 2 among different moments and changes in HH compliance (b) for different moments.** M1: Before patient contact; M2: Before clean/aseptic procedure; M3: After body fluid exposure; M4: After patient contact; M5: After touching patient surroundings. a indicates a significant difference compared to M3 (*P* < 0.001). b indicates a significant difference compared to M3 (*P* < 0.05). c indicates a significant difference compared to M5 (*P* < 0.05).

The incidence of hand-rubbing duration >15 s during Phase 1 were generally higher than in Phase 2, although these differences were not statistically significant (S4a Fig). Similarly, the incidence was higher in soup & water group compared to the ABHR group without statistically significant (S4b Fig). Detailed data can be found in S6 Table.

## Correlation between HH compliance and HAIs incidence

Fig 4 illustrates monthly HH compliance and incidence of HAIs among two phases. The incidence of HAI rose notably from 16.79‰ (95% CI: 16.20‰ to 17.39‰) to 18.71‰ (95% CI: 18.20‰ to 19.24‰) (*P* < 0.001) from phase 1 to phase 2.

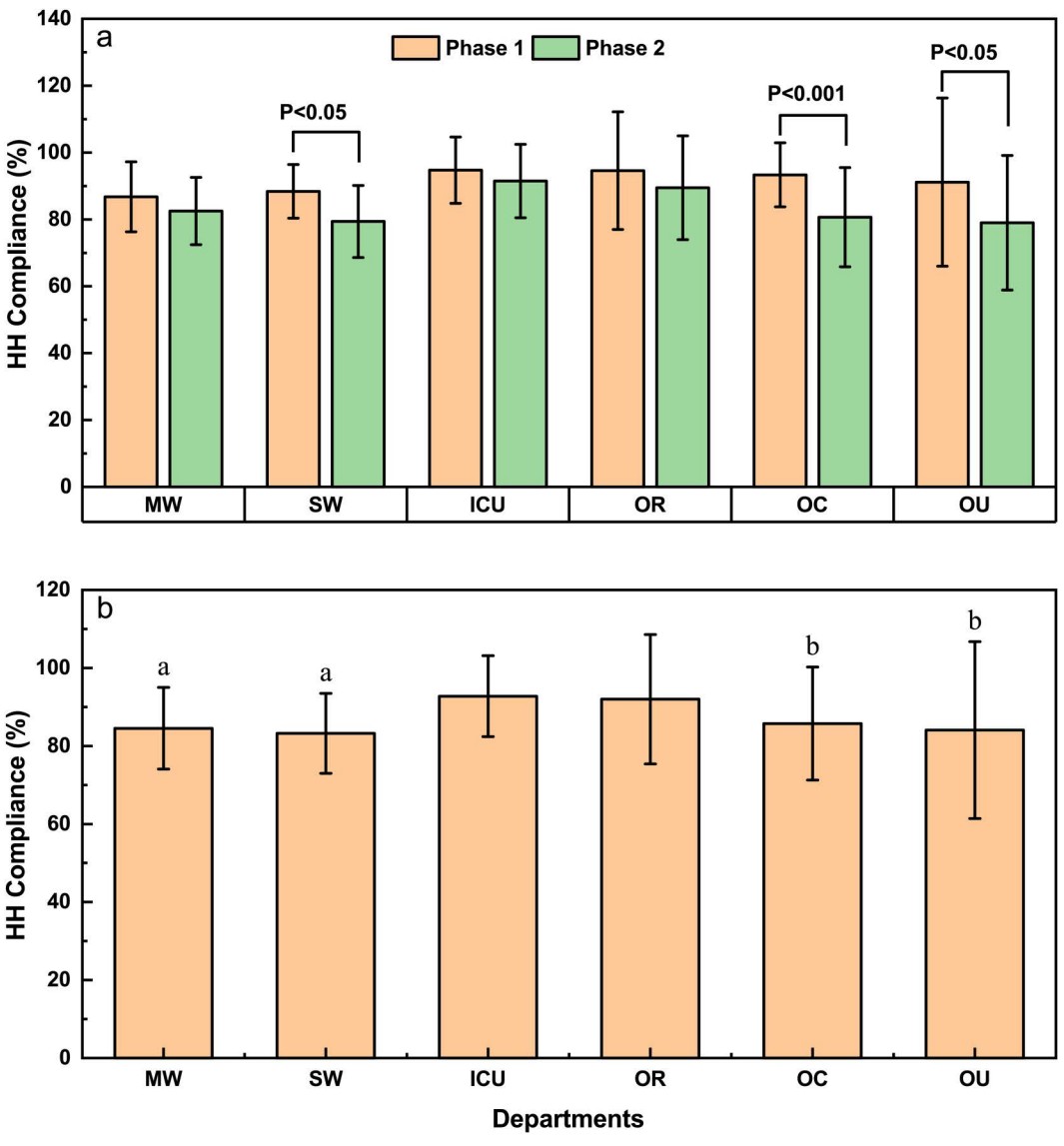

**Fig 3. Comparison of HH compliance (a) between Phase 1 and Phase 2 among different departments and Changes in HH compliance (b) for different departments.** MW: Medical wards; SW: Surgical wards; OP: Operating room; OC: Outpatient clinics; OU: Other units. a indicates a significant difference compared to ICU ($P<0.001$). b indicates a significant difference compared to ICU ($P<0.05$).

Besides, A significant increase was evident in Phase 2 for healthcare-associated lower respiratory tract infections (LRTI) ($P<0.001$), upper respiratory infections (URI) ($P<0.001$), and infections of the eye, ear, nose, throat, or mouth (EENT) ($P<0.001$), however, no significant increase in other HAI sites was observed. Detailed HAI data by infection sites can be found in S7 Table.

Between phase 1 and phase 2, a marked decline in HH adherence was observed, coincident with a significant increase in HAI rates ($p<0.001$). The Pearson's regression model reveals a modest negative association between monthly HH compliance and HAI incidence ($rs=-0.296$), without statistical significance ($P=0.0897$).

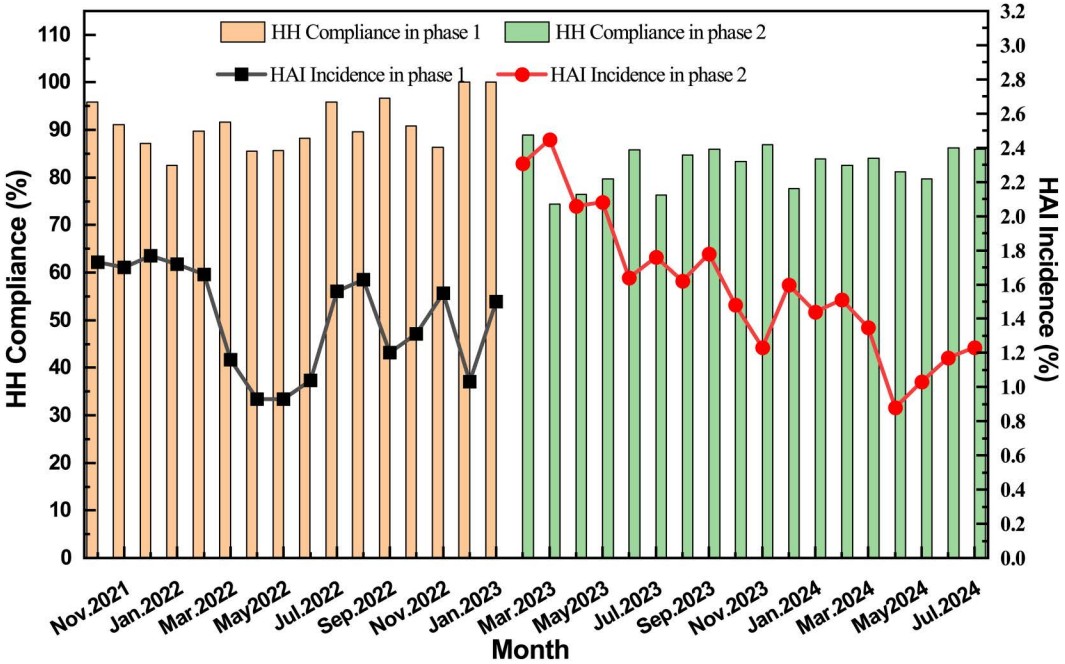

**Fig 4. Monthly HH compliance and incidence of HAIs among two phases.**

## Discussion

### Key findings

The HH compliance decreased from 90.27% to 82.56% notably, while the incidence of HAI increased significantly over the study period. Previous studies have reported self-protection as a major driver of HH among HCWs [20,21]. The pandemic allowed people to change behavior through reactive stimuli such as fear and knowledge, which led to greater compliance with HH [22]. The downward trend of HH in this study could have been driven by reduced public media reporting of the pandemic and resulting lower concern about this threat. This suggests that additional strategies are needed to maintain the high levels of HH achieved during the pandemic. Before the pandemic, our hospital implemented a range of measures, in line with WHO's multimodal strategy, to boost HH compliance among HCWs. These measures included establishing a comprehensive HH policy, supplying various types of hand sanitizers, installing automatic-dispensing hand sanitizer and non-touch taps in high-infection-risk departments, placing HH reminders at points-of-care and hand wash/hand rub stations, providing annual training for all HCWs, offering interactive training, game shows, or electronic videos on World Hand Hygiene Day each year, and publishing the overall HH compliance of the hospital quarterly in the Hospital Infection Management Committee and the infection Control bulletins. This fostered a strong HH awareness among hospital staff. During the pandemic, rigorous prevention and control tactics further solidified a culture of compliance which fostered a sustained culture of compliance. After downgrade in COVID-19 management, HH guidelines established before and during the pandemic are still being adhered to, and the culture of HH compliance has persisted. Consequently, even with a decline in HH compliance following the downgrade in COVID-19 management, the levels remain high relative to other studies [13,23].

The HH compliance of nurses and doctors declined notably between two phases. They involve in medical practice substantially, leading to a higher frequency of HH opportunities observed compared to other staff members. Consequently, their data provides a more precise reflection of the shifts in HH compliance between two phases. Like others [24,25], we

did observe higher compliance for nurses than for doctors and other staff. This likely stems from nurses' high-frequency patient interactions (accounting for 67.40% of the total observed opportunities.), which repeatedly expose them to HH-critical moments, thereby reinforcing their compliance via habit formation.

This is one of the studies to use all of the WHO "5 Moments" to observe HH practice. The HH compliance for M4 decreased significantly from phase 1 to phase 2. In addition to this, the HH compliance was found to be higher for M5 (89.94%) and M4 (85.53%). This finding is similar to other studies that found higher compliance after patient contact [26,27], suggesting that HCWs are acutely aware of the risks of infection acquisition from patients and the healthcare setting and are more focus on personal protection. The finding of highest HH compliance after contact with the patient environment is in contrast to other studies where HH compliance was higher for other moments; for example, increased compliance for prior to a clean or aseptic procedure was documented [7,28], and for after contact with body fluid [11,26,29]. However, in our study, the HH compliance was the lowest after contact with body fluid. This may cause by the interaction between HH and glove use. Gloves were more likely used when performing procedures that may involve exposure to body fluids, and some HCWs assumed that wearing gloves could reliably replace HH. After excluding cases where gloves were worn, the HH compliance increased from 66.32% to 87.50% after contact with body fluid. Previous research indicates glove use can reduce compliance with HH guidelines [30,31], as HCWs usually hold the beliefs that gloves provide enough protection [32], which would help to explain varied findings.

Consistent with previous research [33,34], the HH compliance of ICU is higher versus other departments. This is likely because the ICUs are inherently a high-risk area for HAI with patients usually having unstable immunological conditions and using invasive devices. HCWs in ICU usually place a high emphasis on the prevention and control of HAI, ensuring that infection control measures such as HH are well implemented.

Seasonal fluctuations in infectious diseases may have led to changes in HCWs' HH compliance. Prior research reveals that infectious diseases are more likely to peak in autumn and winter in mainland China [35]. Besides, in southern China, where the hospital is located, the incidence of intestinal infectious diseases surges during autumn [36,37]. Increased infections in autumn could elevate HCWs' risk perception, thus resulting in the highest HH compliance in autumn.

As indicated by prior research [38,39], the duration of hand rubbing is significantly correlated with bacterial counts on hands. We assessed compliance with the 15-s hand-rubbing guideline, as recommended by the National Health Commission of China and implemented on June 1, 2020 [40]. The overall compliance with hand-rubbing duration>15 s is relatively high (91.97%), with a slight decrease observed between the phase 1 and phase 2.

In the present study, HAI incidence increased significantly (from16.79‰ to 18.71‰) when HH compliance decreased from 90.27% to 82.56%. However, we did not find a clear correlation between monthly HHC and HAI. This indicates that several factors other than changes in HH compliance explained the increase. Furthermore, our study shows that LRTI, URI, and EENT increased significantly from phase 1 to phase 2. It is reasonable to speculate that the increase may be related to reduced media coverage, and the resulting decrease in face masking, physical distancing, eye protection and other measures which meant to prevent transmission of COVID-19 [41].

To the best of our knowledge, this study is one of the few studies to conduct the HH compliance and HAIs as the threat of the pandemic begins to subside. During the COVID-19 pandemic, HH and PPE proved essential measures to avoid the spread of SARS-CoV-2 among patients and HCWs [42]. Studies suggest a marked increase in HH compliance during the pandemic [42–44]. Nevertheless, the majority of studies indicate that high level of compliance observed during the pandemic was not sustained; it began to decline progressively soon after [6,43]. Moreover, the duration of the positive effect on HH and subsequent impact on HAIs remain uncertain. To ensure high standards of patient and HCW safety, it is important to know when the effect wears off so as to introduce new strategies on time. Our study conducted an in-depth analysis of the changes in HH between two phases and explored the correlation between HH and HAIs.

### Strengths

The study has potential strengths. First, direct observations of HH compliance and semi-automated surveillance of HAIs were routinely documented in real-world setting on a monthly basis by groups, departments, and moments. Compared with other methods for HH monitoring, direct observation is still considered gold standard [45]. Second, we meticulously tracked all HAIs, classifying them based on criteria provided by the Health Ministry of China [19]. While most previous studies have concentrated on overall HAI or device-associated infections within the ICU or across the hospital, our approach prospectively identified a comprehensive range of potential HAIs correlated with HH. Third, we analyzed the incidence of hand-rubbing duration exceeding 15 s, which has not been universally reported in previous studies.

### Limitations

While the study provides valuable insights, it is not without limitations. Firstly, it is a single-center study conducted within a tertiary hospital with three campuses in China, which may limit the generalizability of the findings to other hospital types or regions. Secondly, this study employed direct observation to collect data on HH, which may have introduced observer bias due to the Hawthorne effect, potentially leading to an increase in reported HH compliance. To minimize potential bias, all observers had additional tasks to perform, ensuring that HCWs were unaware that HH was the focus of the observation. This approach likely reduced the impact of the Hawthorne effect, thereby ensuring a more accurate assessment of HH practices. Thirdly, our study did not examine HH compliance prior to phase 1, particularly during the early stages of the pandemic. Collecting infection prevention data at that time was particularly challenging due to the redirection of resources towards managing the COVID-19 outbreak. Therefore, only the period of phase 1 and phase 2 were monitored to provide insight on the evolution of HH and its correlation with HAIs. Additionally, our study utilized the incidence of HAIs per 100 patients as per the standard established by the Health Ministry of China for HAIs surveillance. This differs from the majority of English-language studies, which employ the rate of HAIs per 1000 patient-days. Lastly, two aspects require further exploration: first, seasonal impact on HCWs' HH compliance, particularly the types of inpatients during different seasons and their influence on HCWs' HH practices. Second, the behavioral mechanisms underlying post-glove-use HH failures, especially false security perceptions.

### Conclusion

This study has revealed the necessity for additional strategies to sustain the high standards of HH compliance achieved during the pandemic. It suggests that HCWs may prioritize self-protection, which could be integrated into future interventions to enhance HH practices. Although there is an observed increase in HAI incidence alongside a decrease in HH compliance, the correlation between them needs further investigation and clarification.

### Supporting information

**S1 Table. Observation values and compliance of two phases.**
(PDF)

**S1 Fig. Change in overall HH compliance between Phase 1 and Phase 2.**
(PDF)

**S2 Table. Observation values and compliance for different healthcare worker groups.**
(PDF)

**S3 Table. Observation values and compliance concerning "Your five moments for hand hygiene" (WHO).**
(PDF)

**S2 Fig. Changes in HH compliance for different moments after the instances of glove use being excluded during M3 (referred to as M3-1).**
(PDF)

**S4 Table. Observation values and compliance of different departments.**
(PDF)

**S3 Fig. Comparison of HH compliance (a) between Phase 1and Phase 2 among different department seasons and Changes in HH compliance (b) for different seasons.**
(PDF)

**S5 Table. Observation values and compliance of different seasons.**
(PDF)

**S6 Table. Observation values and incidence of Hand-rubbing time>15 s.**
(PDF)

**S4 Fig. Incidence of hand-rubbing duration >15 s (a) for different phases and (b) different methods (ABHR and soup &water).**
(PDF)

**S7 Table. Distribution of HAIs in two phases.**
(PDF)

## Acknowledgments

The authors express their sincere gratitude to the entire team at the Infection Prevention and Control Department of The Central Hospital of Wuhan, which is affiliated with Tongji Medical College, Huazhong University of Science and Technology. We are particularly thankful for their diligent efforts in observing and meticulously documenting HH practices, which were instrumental to our study.

## Author contributions

**Conceptualization:** Jinqi Wang, Yaofei Liu, Neng Li, Yifen Zhu, Min Du, Xiaoman He.

**Data curation:** Jinqi Wang, Neng Li, Yifen Zhu, Min Du.

**Formal analysis:** Jinqi Wang, Yaofei Liu, Xiaoman He.

**Funding acquisition:** Jinqi Wang, Neng Li, Xiaoman He.

**Investigation:** Jinqi Wang.

**Methodology:** Jinqi Wang, Yaofei Liu, Xiaoman He.

**Project administration:** Jinqi Wang, Xiaoman He.

**Resources:** Neng Li, Xiaoman He.

**Software:** Jinqi Wang, Neng Li, Xiaoman He.

**Supervision:** Jinqi Wang, Xiaoman He.

**Validation:** Yaofei Liu, Neng Li, Yifen Zhu, Min Du.

**Visualization:** Jinqi Wang, Xiaoman He.

**Writing – original draft:** Jinqi Wang.

**Writing – review & editing:** Jinqi Wang, Yaofei Liu, Neng Li, Yifen Zhu, Min Du, Xiaoman He.

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
