## [Decision Letter · Decision Letter 0]

29 Jun 2025

PONE-D-25-21668
Hand hygiene after the COVID-19 pandemic: Is it still at a high level?
PLOS ONE

Dear Dr. He,

Thank you for submitting your manuscript to PLOS ONE. After careful consideration, we feel that it has merit but does not fully meet PLOS ONE’s publication criteria as it currently stands. Therefore, we invite you to submit a revised version of the manuscript that addresses the points raised during the review process.

We look forward to receiving your revised manuscript.

Kind regards,

Mona Nabulsi, MD, MS

Academic Editor

PLOS ONE

**Journal Requirements:**

1. When submitting your revision, we need you to address these additional requirements.
 
Please ensure that your manuscript meets PLOS ONE's style requirements, including those for file naming. The PLOS ONE style templates can be found at 
https://journals.plos.org/plosone/s/file?id=wjVg/PLOSOne_formatting_sample_main_body.pdf and 
https://journals.plos.org/plosone/s/file?id=ba62/PLOSOne_formatting_sample_title_authors_affiliations.pdf
 
2. Thank you for stating the following financial disclosure: 
This study was funded by Wuhan Science Foundation for Youths [Grant number: WG21Q02].  
 
Please state what role the funders took in the study.  If the funders had no role, please state: "The funders had no role in study design, data collection and analysis, decision to publish, or preparation of the manuscript." 
If this statement is not correct you must amend it as needed. 
Please include this amended Role of Funder statement in your cover letter; we will change the online submission form on your behalf.
 
3. When completing the data availability statement of the submission form, you indicated that you will make your data available on acceptance. We strongly recommend all authors decide on a data sharing plan before acceptance, as the process can be lengthy and hold up publication timelines. Please note that, though access restrictions are acceptable now, your entire data will need to be made freely accessible if your manuscript is accepted for publication. This policy applies to all data except where public deposition would breach compliance with the protocol approved by your research ethics board. If you are unable to adhere to our open data policy, please kindly revise your statement to explain your reasoning and we will seek the editor's input on an exemption. Please be assured that, once you have provided your new statement, the assessment of your exemption will not hold up the peer review process.
 
4. Please include captions for your Supporting Information files at the end of your manuscript, and update any in-text citations to match accordingly. Please see our Supporting Information guidelines for more information: http://journals.plos.org/plosone/s/supporting-information.

Reviewers' comments:

Reviewer's Responses to Questions

**Comments to the Author**

1. Is the manuscript technically sound, and do the data support the conclusions?

Reviewer #1: Yes

Reviewer #2: Yes

2. Has the statistical analysis been performed appropriately and rigorously? 

Reviewer #1: Yes

Reviewer #2: Yes

3. Have the authors made all data underlying the findings in their manuscript fully available?

Reviewer #1: Yes

Reviewer #2: No

4. Is the manuscript presented in an intelligible fashion and written in standard English?

Reviewer #1: Yes

Reviewer #2: Yes

5. Review Comments to the Author

**Reviewer #1: **Thank you for the opportunity to review this timely and interesting manuscript. In my opinion, it has many strengths, but in order to be suitable for publication, it still requires further development. Below are my detailed comments.

e.g. line 51: An abbreviation must be spelled out in the main text, even if it has already been defined in the abstract (e.g., HAI, HCW).

e.g. line 58: Percentage values should be rounded consistently throughout the article, for example 91.76% = 91.8%

Methods:

lines 87–89: The meaning of this statement remains unclear, particularly the reference to categories A and B.

lines 110–121: The methods section should more clearly describe who conducted the observations, how they were carried out in practice, and how the individuals being observed were selected. Was it possible that the same individual was observed multiple times? Were the individuals informed afterward that they had been observed? How were different professional groups distributed across the various hospital departments? What was the rationale for selecting the specific time points for observation?

Results:

Lines 177–180: It is unclear how the information was obtained regarding the specific event to which the HH instance was related.

Figure 2: Figures and their legends should not use abbreviations alone. Abbreviations such as M1, M2, etc., should be explained in full.

Figure 4 and the accompanying text: The y-axis should start at zero. How might factors such as the seasons and the differing lengths of observation periods have influenced the results? Are there differences in the types of patients hospitalized during different seasons?

Discussion:

Lines 250–252: “Consequently, even with a decline in HH compliance following the downgrade in Covid-19 management, the levels remain high relative to other studies.” -> It would be valuable to consider potential explanations for this finding. Could it also be due to differences in study methods or hospital environments?

Lines 256–257: It would also be important to consider the reasons behind the differences observed between professional groups.

Lines 268–270: “Gloves were more likely used when performing procedures that may involve exposure to body fluids, and some HCWs assumed that wearing gloves could reliably replace HH.” What methods were used to determine that in this hospital healthcare workers assumed glove use could reliably substitute for hand hygiene?

Abstract:

As the authors themselves note in the Discussion, there may be multiple underlying reasons for changes in hand hygiene and healthcare-associated infections. Therefore, in my opinion, the correlation analysis is given too prominent a role in the Abstract.

**Reviewer #2:** This is an interesting and well written manuscript and it it illustrates the decline of hand hygiene compliance that occurred after the end of the pandemic in one hospital. No clear correlation with HAIs and HH rates in this study.

6. PLOS authors have the option to publish the peer review history of their article (what does this mean?). If published, this will include your full peer review and any attached files.

Reviewer #1: No

Reviewer #2: No

---

## [Author Response · Author response to Decision Letter 1]

24 Jul 2025

PONE-D-25-21668

Hand hygiene after the COVID-19 pandemic: Is it still at a high level?

Dear Reviewer,

We are grateful for your careful reading of our manuscript and the constructive feedback provided. Your insights have helped us improve the clarity and rigor of our study.

 Sincerely,

 Xiaoman He

 2025-7-24

Point-by-Point Response to Reviewer 1

1. Comment: line 51: An abbreviation must be spelled out in the main text, even if it has already been defined in the abstract (e.g., HAI, HCW).

 Response：We have carefully revised the manuscript to ensure that all abbreviations are spelled out at their first occurrence in the main text, even if previously defined in the abstract. We have added the full terms for HAI (Healthcare-Associated Infection) at line 53, HCWs (Healthcare Workers) at line 75, and HH (Hand hygiene) at line 59 in the Revised Manuscript, highlighted in red for easy review.

2. Comment: line 58: Percentage values should be rounded consistently throughout the article, for example 91.76% = 91.8%.

 Response：Throughout the main text, all percentage values now uniformly display two decimal places. For cited data in the Introduction, values originally with one decimal (21.5%, 75.1%) were standardized to 21.50% and 75.10% (Line 64), whole numbers (74%, 5%, 89%) were formatted as 74.00%, 5.00%, and 89.00% (Lines 77-78).

This ensures numerical consistency while preserving original values. Modifications are highlighted in red in the Revised Manuscript (Lines 64, 77-78).

3. Comment: Methods: lines 87-89: The meaning of this statement remains unclear, particularly the reference to categories A and B.

 Response：We have added a citation to the Law of the People's Republic of China on the Prevention and Treatment of Infectious Diseases in the Revised Manuscript[1]. This legally binding framework classifies infectious diseases based on their severity and social impact into three categories: Category A, B and C. Although initially classified as Category B, COVID-19 was managed with Category A control measures (the strictest level) nationwide since January 2020, as mandated by China's National Health Commission [2]. This exceptional approach ("Category B disease with Category A measures") was implemented due to its pandemic potential.

These clarifications and citations are incorporated in the Methods section (Lines 93-102, highlighted in red) of the Revised Manuscript, with full details in the bibliography.

4. Comment: Methods: lines 110-121: The methods section should more clearly describe who conducted the observations, how they were carried out in practice, and how the individuals being observed were selected. Was it possible that the same individual was observed multiple times? Were the individuals informed afterward that they had been observed? How were different professional groups distributed across the various hospital departments? What was the rationale for selecting the specific time points for observation?

 Response：1. Identification of Observers: As stated in line 113 of the original manuscript (line 134 of the Revised Manuscript), the observers were twelve staff members from the Infection Prevention and Control Department. They possessed proficiency in the WHO's "Five Moments for Hand Hygiene" and had undergone training via case scenarios and instructional videos to ensure standardized observation criteria. Besides, the twelve observers were divided into six fixed pairs, with each pair assigned exclusively to one of the six stratum categories (medical wards, surgical wards, operating rooms, outpatient clinics, ICUs, other units).

2. Observation implementation: In each observational session, a pair of observers independently assessed a real-life care situation, monitoring the same HCW's HH practices and care sequence. They recorded their findings on separate paper forms, indicating the observed individual's profession, the specific HH moment, the action taken, and glove usage. These forms were later compared, with discrepancies resolved through discussion to ensure the accuracy of the observations. To mitigate the Hawthorne effect, observations were seamlessly integrated into observers’ routine duties (e.g., ward rounds), ensuring HCWs remained unaware that their HH practices were under evaluation. Observers discreetly recorded HH opportunities on the WHO observation paper forms during 20-30 minute sessions, mimicking routine ward activities.

3. Participant Selection: Stratified random sampling was employed to select HCWs for observation. Departments in the hospital were stratified into six categories, including surgical wards, medical wards, operating rooms, outpatient clinics, ICUs, and other units (including radiology, laboratory, rehabilitation and other supporting departments). The proportion of total observation sessions was allocated to each stratum based on their characteristics and risks of HAIs: medical wards (15%), surgical wards (15%), operating rooms (15%), outpatient clinics (20%), ICUs (20%), other units (15%). Within each stratum, the pre-assigned pair of observers for the stratum recorded HH opportunities encountered at random among multiple HCWs engaged in routine care during the allocated time slots. This approach ensured a representative sample of HCWs across different departments.

4. Repeat Observations: The study encompassed three campuses of our hospital (all large general hospitals), covering over 4,000 HCWs. To maintain objectivity, observation protocols specified: (1) no HCW was observed more than once within the same session, and (2) HCWs observed within the previous week were excluded from subsequent observations. Although repeated observation was unavoidable during the 34-month-long period, the probability remained negligible due to these protocols, the large sample size, and multi-campus design.

5. Post-Observation Notification: Per ethics waiver (2021-016), no individual feedback or intervention was provided to observed HCWs during or after the observations. However, aggregate HH compliance rates of the hospital were publicly reported quarterly through the Hospital Infection Management Committee briefing and Infection Control Bulletin to maintain transparency.

6. Distribution of Professional Groups: In our study, the professional groups were categorized into doctors, nurses, paramedical staff, and cleaning staff. The distribution of these groups across the hospital departments was based on the specific roles and responsibilities required in each department. Based on 2233 HH opportunities, nurses accounted for 75.15%, physicians 18.09%, paramedical staff 4.84%, and cleaners 1.92%.

To provide more details: Doctors and nurses were primarily distributed across clinical departments such as medical wards, Surgical wards, Operating room, Outpatient clinics, and ICUs. Paramedical staff included technicians and assistants who supported diagnostic and therapeutic procedures. They were mainly located in departments such as radiology, laboratory, and rehabilitation. Cleaning staff were responsible for maintaining the cleanliness and hygiene of the hospital environment and were distributed across all departments and wards.

7.Rationale for Timing: Most observations were conducted during morning care (7:30-10:30) with each observation session lasting 20-30 minutes. This period is characterized by high-intensity care activities, allowing each session to maximize the collection of diverse HH opportunities while reflecting real-world practice under typical workload pressures, in line with WHO-recommended dynamic scenario monitoring.

The Revised Manuscript has been updated to elaborate on the Methods section (lines 89-90, 132-168), specifically addressing the identification of observers, observation protocols, participant selection, repeat observation controls, and related aspects. These revisions are clearly marked in red for ease of review.

5. Comment: Results: Lines 177-180: It is unclear how the information was obtained regarding the specific event to which the HH instance was related.

Response：The "Five Moments for Hand Hygiene" concept was first systematically proposed in the Hand Hygiene Technical Reference Manual released by the World Health Organization (WHO) in 2009 [3]. This framework summarizes the key timings for HCWs to perform HH into five moments, which has become a core standard for infection control in the global healthcare sector. These Five Moments include: Prior to patient contact, Prior to a clean or aseptic procedure, After contact with body fluid , After patient contact, and After contact with the patient environment.

In this study, the WHO observation form containing these five moments was translated into Chinese. Each HH event was classified into a specific moment was determined through direct, real-time observation of healthcare activities by observers who were proficient in the WHO's "5 Moments for Hand Hygiene (HH)" and trained via case scenarios and instructional videos, with the entire process strictly aligned with the WHO framework. To ensure accuracy, two observers independently classified each HH opportunity during the same care sequence and discordant classifications were resolved through consensus discussion, in accordance with the requirements outlined in the WHO Hand Hygiene Technical Reference Manual .

 The observation process and requirements aligns with the guidelines outlined in the validated WHO Hand Hygiene Technical Reference Manual. This methodological detail has been clarified in the Revised Manuscript (Lines 119-123). Additionally, we have included the WHO Technical Reference Manual as reference at the first mention of the Five Moments (Lines 120).

6. Comment: Figure 2: Figures and their legends should not use abbreviations alone. Abbreviations such as M1, M2, etc., should be explained in full.

Response：In the Revised Manuscript, we have revised the Figure 2 legend to include full explanations of abbreviations. The revised legend (Lines 245-247) is highlighted in red for easy reference.

7. Comment: Figure 4 and the accompanying text: The y-axis should start at zero. How might factors such as the seasons and the differing lengths of observation periods have influenced the results? Are there differences in the types of patients hospitalized during different seasons?

Response：1. As suggested, the y-axis of Figure 4 has been revised to start at zero, ensuring a more appropriate representation of the data. The updated version is provided as Revised Figure 4.

2.Observation period consistency: All observations were strictly limited to 20-30 minutes per session across all seasons. This standardized duration minimizes variability in data density.

3.Seasonal effects on HH compliance: As per the World Meteorological Organization (WMO) definition, seasons were demarcated as: Spring: March–May; Summer: June–August; Autumn: September–November; Winter: December–February. We conducted an in-depth analysis of HH compliance across four seasons and found that there were significant differences in HH compliance among different seasons, with HH compliance in autumn being significantly higher than in spring (P < 0.001), summer (P < 0.05), and winter (P < 0.05).

Potential drivers of this variation may relate to seasonal fluctuations in disease patterns. Prior research reveals that infectious diseases are more likely to peak in autumn and winter in mainland China [4]. Besides, in southern China, where the hospital is located, the incidence of intestinal infectious diseases surges during autumn [5,6]. Increased infections in autumn could heighten awareness of infection control among HCWs, indirectly impacting HH compliance.

This finding corroborates the reviewer's insight regarding seasonal influences. We have now integrated in-depth seasonal analyses into the Revised Manuscript, including in the Results section (Lines 268-275) and Discussion section (Lines 364-369). Additionally, the corresponding figure (S3 Fig) and table (S5 Table) for the seasonal analysis have been added to the supporting information. Furthermore, we plan to conduct deeper investigations into seasonal factors, such as performing department-stratified analyses to clarify the types of inpatients during different seasons and their influence on HCWs’ HH practices in future (Lines 428-431 in Limitations section).

8. Comment: Discussion: Lines 250-252: “Consequently, even with a decline in HH compliance following the downgrade in Covid-19 management, the levels remain high relative to other studies.” -> It would be valuable to consider potential explanations for this finding. Could it also be due to differences in study methods or hospital environments?

Response：1. Study methodology consistency: Consistent with the implementation conditions of other researches[7], our study employed direct observation to collect data on HH, which may have introduced observer bias due to the Hawthorne effect, potentially leading to an increase in reported HH compliance. To minimize potential bias, all observers had additional tasks to perform, ensuring that HCWs were unaware that HH was the focus of the observation. This approach likely reduced the impact of the Hawthorne effect, thereby ensuring a more accurate assessment of HH practices. Compared with other studies that employed direct observation, the levels of HH compliance remained high even after the downgrade in COVID-19 management[7,8].

Besides, Our observation period (20–30 minutes per session) and compliance assessment criteria have been standardized and aligned with the WHO Hand Hygiene Technical Reference Manual [3], thus avoiding discrepancies in results caused by differences in research methods.

Therefore, we have reason to believe that the high levels of HH compliance are not strongly related to our research methodology.

2.Hospital-specific culture and long-term practice: As a large tertiary hospital with three campuses, our patient population includes a high proportion of immunocompromised individuals and complex cases, requiring ongoing vigilance in infection prevention—independent of pandemic status. Thus, prior to the COVID-19 pandemic, in accordance with the WHO's multimodal strategy for HH improvement, our hospital maintained a variety of measures to enhance HCWs’ HH compliance. These included establishing a comprehensive HH policy, supplying various types of hand sanitizers, installing automatic-dispensing hand sanitizer and non-touch taps in high-infection-risk departments, placing HH reminders at points-of-care and hand wash/hand rub stations, providing annual training for all HCWs, offering interactive training, game shows, or electronic videos on World Hand Hygiene Day each year, and publishing the overall HH compliance of the hospital quarterly in the Hospital Infection Management Committee and the infection Control bulletins. Through these measures, the HCWs had developed a strong awareness of HH.

During the COVID-19 pandemic, our hospital enacted rigorous preventive and control tactics (Method section, lines 102-105, and Table 1 ), which fostered a sustained culture of compliance. During Phase 2 (management for Covid-19 downgrade), HH guidelines established before and during the pandemic are still being adhered to, with no emphasis on implementing additional active improvement measures, and the culture of HH compliance has persisted.

Therefore, even with a decline in HH compliance following the downgrade in COVID-19 management (from 90.27% to 82.56%), the levels remain comparatively high relative to other studies [7,8].

The Revised Manuscript has been updated to explain the hospital’s efforts regarding HH prior to the pandemic elaborate in the Discussion section (lines 317-331), with revisions highlighted in red for ease of review.

9. Comment: Discussion: Lines 256-257: It would also be important to consider the reasons behind the differences observed between professional groups.

Response：Similar to other studies [9, 10], our research observed that nurses had far more HH opportunities than other professional groups, accounting for 67.40% of the total observed opportunities. Nurses typically engage in more frequent direct patient

---

## [Decision Letter · Decision Letter 1]

3 Sep 2025

Hand hygiene after the COVID-19 pandemic: Is it still at a high level?

PONE-D-25-21668R1

Dear Dr. He,

We’re pleased to inform you that your manuscript has been judged scientifically suitable for publication and will be formally accepted for publication once it meets all outstanding technical requirements.

Kind regards,

Mona Nabulsi, MD, MS

Academic Editor

PLOS ONE

Reviewer's Responses to Questions

**Comments to the Author**

1. If the authors have adequately addressed your comments raised in a previous round of review and you feel that this manuscript is now acceptable for publication, you may indicate that here to bypass the “Comments to the Author” section, enter your conflict of interest statement in the “Confidential to Editor” section, and submit your "Accept" recommendation.

Reviewer #1: (No Response)

2. Is the manuscript technically sound, and do the data support the conclusions?

Reviewer #1: Yes

3. Has the statistical analysis been performed appropriately and rigorously? 

Reviewer #1: Yes

4. Have the authors made all data underlying the findings in their manuscript fully available?

Reviewer #1: Yes

5. Is the manuscript presented in an intelligible fashion and written in standard English?

Reviewer #1: Yes

6. Review Comments to the Author

Reviewer #1: Thank you for the opportunity to re-evaluate this interesting manuscript. In my opinion, it now meets the requirements for publication. However, I would like to leave one final comment, which I hope the authors will consider.

In my opinion, all numerical values should be rounded to one decimal place or none, both in the abstract and the main text. Typically, it is not necessary to present two decimal places in studies of this kind.

For example, in lines 75–78:

“Seven studies with 2,377 health care providers reporting the estimated overall HH compliance during the pandemic was 74.00%, which was higher than that reported in pre-pandemic studies (5.00%–89.00%) [11].”

- Decimal places are not needed here.

And for example, in lines 204–206:

“A total of 2,233 HH opportunities were observed during two phases, and HCWs conducted HH according to recommendations in 1,918 of the occasions, yielding a total compliance rate of 85.89% (95% CI: 84.38% to 87.31%).”- One decimal place is sufficient.

7. PLOS authors have the option to publish the peer review history of their article (what does this mean?). If published, this will include your full peer review and any attached files.

Reviewer #1: No

---

## [Editor Report · Acceptance letter]

PONE-D-25-21668R1

PLOS ONE

Dear Dr. He,

I'm pleased to inform you that your manuscript has been deemed suitable for publication in PLOS ONE. Congratulations! Your manuscript is now being handed over to our production team.

Kind regards,

on behalf of

Dr. Mona Nabulsi

Academic Editor

PLOS ONE